# Using Social Norms to Change Behavior and Increase Sustainability in the Real World: A Systematic Review of the Literature

**Paulius Yamin [1,2,3,*], Maria Fei [4], Saadi Lahlou [1,5] and Sara Levy [6]**

[1] Department of Psychological and Behavioural Science, London School of Economics and Political Science, London WC2A 2AE, UK
[2] Faculty of Transport Engineering, Vilnius Gediminas Technical University, 10223 Vilnius, Lithuania
[3] Behavioural Lab LT, 14184 Vilnius, Lithuania
[4] Behavioural Science Lead, Ipsos Healthcare, London E1W 1WY, UK
[5] Paris Institute for Advanced Study, 75004 Paris, France
[6] EMEA Brand and Consumer Insights, Google, London SW1W 9QT, UK
**\*** Correspondence: p.yamin-slotkus@lse.ac.uk; Tel.: +370-695-554-07

**Abstract:** Behavioral change interventions based on social norms have proven to be a popular and cost-effective way in which both researchers and practitioners attempt to transform behavior in order to increase environmental and social sustainability in real-world contexts. In this paper, we present a systematic review of over 90 empirical studies that have applied behavioral change interventions based on social norms in field settings. Building on previous research about the sources of information that people use to understand social norms and other local determinants of behavior, we propose a framework organized along two axes that describe intervention context (situated interventions applied in the same context where the target behavior happens versus remote interventions that are applied away from that context) and type of normative information leveraged (interventions that provide summary information about a group versus interventions that expose participants to the opinions and behaviors of others). We also illustrate successful applications for each dimension, as well as the social, psychological and physical determinants of behavior that were leveraged to support change. Finally, based on our results, we discuss some of the elements and practical mechanisms that can be used by both researchers and practitioners to design more integral, effective and sustainable social norm intervention in the real world.

**Keywords:** social norms; social influence; sustainability; installation theory; normative perceptions; behavioral change; intervention; systematic review; field studies

## 1. Introduction

Administrators, policymakers and practitioners are increasingly aware of how transforming individual and collective trends of behavior is essential to achieve their goals [1–3]. Tackling the challenges in environmental and social sustainability that our planet currently faces cannot be done without important and long-term transformations on the everyday behaviors of individuals: "Desirable goals, such as lowering greenhouse gas emissions, reducing waste, and increasing energy and water efficiency can be met only if high levels of public participation are achieved" ([4], p. 544).

Such behavioral changes do not occur spontaneously as policies are established; therefore "behavioral change" must be designed. Because of this, even the best meaning and most sophisticated management systems, policies, laws or programs require some behavioral change in their targets to succeed, and can often fail if these changes do not happen [5–7]. There are many theories of behavior and of behavioral change [1,8]; here we shall focus on one of the most popular, the use of social norms.

Influencing social norms, or the rules that describe what a certain reference group considers to be typical or desirable behavior in certain contexts and situations [9], is a popular way in which researchers and practitioners attempt to transform behavior in real-world contexts [3,8,10–13]. These initiatives have often focused on behavioral changes that increase the environmental and social sustainability of their target contexts. Successful examples include both challenges directly related to environmental issues such as energy and water consumption [14,15], sustainable transport use [16,17], recycling [18,19] and food choices [20–22], as well as broader social sustainability issues such as alcohol consumption [23], hygiene [24,25], and harassment and violence [6,26].

In these and other realms, since "humans are especially motivated to understand and to follow the norms of groups that we belong to and care about" ([3], p. 184), changing the perceptions we have about social norms is a powerful and cost-effective way of creating or reinforcing collective changes in behavior. Furthermore, since social norms are based on mutual regulation between social actors [6,7], it can also reduce the need for the costly authoritative enforcement or attitudinal change that many change initiatives attempt by using the "vigilante effect" through which members of a society tend, in situation, to enforce "correct" behavior in others ([27], pp. 140–144). The power of social norms in behavioral change does not come only from the natural inclination to imitate others or from the necessity to know what is appropriate to do in a given situation, but it is also rooted in the human desire to belong to one's community. As Kurt Lewin aptly noted in his seminal paper on behavioral change, humans fear to stand out from the group: "the unwillingness of the individual to depart too far from group standards" ([28], p. 273) is a strong lever to influence behavior.

Social norms have been a central topic in psychological research for a long time [29–32]. Researchers in this and other fields have found relevant behavioral effects of interventions based on social norms in a very wide variety of domains, from pro-environmental behaviors [14,16,18,33] to violence and harassment [34–36], and from health related and risk behaviors [24,37–39] to gambling [40,41], to mention just a few examples. By testing some of the mechanisms and contexts in which social norms can have relevant effects, these field studies are an important resource for both practitioners and researchers to design more effective interventions to address all kinds of policy and social challenges.

Recent popular and influential texts that explore the topic of social norms [2,23,34] have described some of the practical mechanisms that have been used to achieve behavioral change in social norm interventions. These include, for example, the Social Norm marketing approach, the Personalized Normative Feedback approach, focus groups discussions [23], media campaigns [2,34], and also legal means, economic incentives, and deliberation [2]. Nevertheless, a systematic exploration of how these and other intervention mechanisms have been applied in the empirical literature was still lacking and could greatly inform the design and research of social norm interventions to increase sustainability in real-world contexts.

In this paper, we present such a review from a wide sample of over 90 empirical studies that have applied behavioral change interventions based on social norms in field settings. Our objective is not to evaluate or compare the reported efficacy of these interventions (which is for us a second, more difficult step and would require different methods and a different sample), but rather to explore how researchers have leveraged these sources of information to influence normative perceptions and behaviors. By doing this, our goal is to present an overview of current practice in a systematic analytic framework that is useful for researchers and practitioners. In this way, we hope to both contribute to a better understanding of how these interventions are applied in the literature, and also to inform the systematic testing and application of more effective and sustainable interventions both inside and outside academic settings.

## 1.1. Social Norms and Behavioral Change

Social norms determine the behaviors that, among all those that are possible in a given situation, "others (as a group, as a community, as a society…) think are the correct ones, for one reason or another" ([27], p. 124). By defining the socially accepted ways of acting in certain contexts and situations, social norms are a central part of social regulation, the process through which "other

stakeholders regulate our activity" ([27], p. 124). By doing this, they also mark our membership and place in a group, how we perceive social situations, how we relate and interact with others, and how we respond to cultural products [29].

Empirical studies in psychology and other disciplines have explored extensively the potential of using interventions based on social norms to transform behavior in real-world settings [3,11,13,23]. These empirical studies have identified a wide variety of general topics of enquiry and moderators that influence the potential of social norms to change behavior. We list a few of the most popular areas of research below, differentiating between those that focus on the importance of the attributes of behaviors and social norms, and those that emphasize the importance of the contexts in which those behaviors and social norms are embedded.

### 1.1.1. The Importance of the Attributes of Social Norms and Behaviors

One of the best-known distinctions in the social norm literature is the one between norms that describe typical behaviors (called descriptive norms in Cialdini et al.'s Focus Theory [42] and in Bicchieri [2]), and those that describe desirable behaviors (called subjective norms in the Theory of Reasoned Action of Fishbein and Ajzen [43], social norms in Bicchieri [2] and injunctive norms in Cialdini et al. [3], see also [44,45] for further details). While the former ("descriptive norm") refers to what I perceive to be typical behavior in a situation [3] (what most people do in Cialdini et al. [42]), the latter ("injunctive norm") refers to what I perceive to be desirable behavior (what most people ought to do in Cialdini at al. [42]).

Several studies have explored the relationship between these two types of norms. There is evidence of descriptive norms having heterogeneous effects depending on the reference behavior levels (which can be desirable or undesirable in the intervention),with "boomerang effects" documented in some cases [18,46–48].This has been successfully counteracted by adding injunctive elements to the messages, especially those reinforcing the desirable behaviors [49]. For example, messages about the high prevalence of petrified wood stealing [50] or about how your household consumes less energy than your neighbors' [51] might lead to increases in undesirable behaviors if injunctive messages (i.e., about social desirability or un-desirability) are not added as well. In another study about voting behaviors, Gerber et al. [52] also found evidence that descriptive messages about the high prevalence of a target behavior (i.e., thousands people vote so you should too) can be more effective to change behavior than those about low prevalence (i.e. a low proportion of people vote so you should do it) or even injunctive norms (i.e., it's the right thing to do or it's a civic duty), especially among people that don't engage often in the behavior. Injunctive norms, on the other hand, seem to be more effective when combined with accuracy or efficiency goals [53], and when formulated in a positive manner (i.e., "people think you should do this") in contexts where descriptive norms are weak [54].

Dolcini et al. [55] have also emphasized the importance of taking into account and measuring some basic attributes of social norms such as norm homogeneity, strength and stability, as well as how these characteristics relate to different clusters in the population [55]. Lapinski & Rimal [45] underline how normative influences are determined by basic attributes of behaviors, such as behavioral ambiguity and privacy [24].

### 1.1.2. The Importance of the Context

The range of activities that people can undertake is not just determined by what is perceived as "socially acceptable". The physical affordances of the environment and the embodied interpretive systems of subjects also play an important role in locally funneling behavior [27]. In this way, when we decide to keep our place in line at the customs control point, we do not do it only because we read a message that says that most people do it as well. We also imitate others and are directly cued and corrected by specialized personnel and other fellow travelers on the spot, we have internalized norms and skills that indicate how to act in that situation, and there are often physical signs, marks and barriers that funnel the expected behavior.

Installation Theory states that behaviors are, in actual situations, the result of a compound of factors, of three types or "layers": local affordances of the material context, embodied competences of the subject, and social control. These components coalesce in situ to provide a scaffolding and constraining behavioral "installation" that funnels behavior [27]. Because of that, effective behavioral interventions should not only address the individual's "internal" factors that could predict behavior (such as representations, competences or perceptions), but also the local installations in which these behaviors are enacted. As a matter of fact, in the generic framework of Installation Theory, which encompasses the widest number of factors, the social norm appears as a limited approach to intervention. Nevertheless, the social norm, as it represents the standard behavior performed in the typical circumstance by the group members ("the way we do around here"), is a handy proxy for adequate behavior that can be reminded to the subjects in interventions and policy implementation. That is why social norms have been instruments of choice for such interventions.

Researchers working on social norm interventions have tested the idea that social norms are context-dependent, situational and require a focus of our attention to affect behavior. Several studies [42,46,49,56] have presented evidence of how the influence of social norms on behavior is not uniform in terms of context or time, but rather depend on them. Other researchers have underlined the importance of the perceptions we have of social norms and the sources we use to gather normative information [3,23,45,57]. As Tankard and Paluck [3] argue, psychologists are concerned, less about the actual rates of behavior of a population (which are more the interest of policymakers), and more about the "community members' subjective perceptions of the norm" ([3], p. 181). These perceptions "become a reality and a guide for their own behavior, even when the perceptions are inaccurate" [3] (p. 183). The authors also distinguish three main sources that individuals use to gather normative information: the behaviors we see in others, the summary information we receive about a group, and the signals that different institutions send [3]. Also, Prentice and Miller [57] have shown how, in practice, a social norm can still prevail and influence behavior even when it is believed to be unpopular and dysfunctional (a phenomenon known as pluralistic ignorance).

When we pay attention to a certain social norm, we often do so in relation to a specific reference group that we think engages in and/or approves a certain behavior (for example our friends, our parents, our co-workers, or our neighbors). Consequently, many researchers that have also focused on how we use different reference groups to interpret social norms [35,58–60], the networks that organize them [35,61] and our sense of identity in relation to them [62,63]. These researchers have presented evidence on how involving local promotors [59,60], high-profile messengers [64], friends [65], role-models [66], in-groups [67] but also outgroups [57], and geographically and demographically close individuals [68], are all related to higher effectivity of social norm interventions.

### 1.2. Our Review

Our review emerged from the need to perform a systematic exploration into the intervention strategies and mechanisms used in behavioral change interventions based on social norms. Unfortunately, many researchers limit themselves to exploring the conceptual and research dimensions of behavioral change interventions without paying much attention to how these ideas and findings can inform and improve the practical design and implementation processes of such interventions. As Davis et al. ([69], p. 2218) argue, "while the social norms approach is based in a rich theory, the theory does little to illuminate implementation details of interventions".

Nevertheless, as already mentioned, there are authors that have explored a limited number of practical mechanisms that have been used in social norm interventions [2,23,34]. Also, in one of the better-known examples outside the realm of social norms, Michie et al. [70,71] have identified 93 behavioral change techniques to report the intervention procedures of any kind of behavioral change intervention. Although relevant and useful, these techniques don't directly consider social regulation and its related mechanisms, but rather, in a more general way, processes such as social support or comparison of behaviors.

Based on a broad systematic sample, our review aims to produce a first overview of some of the main intervention strategies and mechanisms used specifically in field social norm interventions. With this objective in mind, and taking into account the previous literature on the subject, we focus on four areas of enquiry that are particularly relevant for the difficult task of developing intervention strategies once a diagnosis has been conducted [72]. These four areas pertain to:

1) Context of application (situated vs remote). The contexts in which intervention mechanisms are applied relative to the target behavior (specifically, following Lahlou [27], whether they are applied in the context where the target behavior happens or away from it)
2) Type of normative information (group summary information vs exposure to behaviors and opinions). The type of normative information that are intentionally leveraged in the intervention to influence behavior (specifically, following Tankard & Paluck [3], whether interventions rely on group summary information or exposure to the behaviors of others)
3) Intervention mechanisms. The different intervention mechanisms that are used to leverage the physical, psychological and social determinants of behavior (following Lahlou [27])
4) Combination of mechanisms. How the previous three elements are combined in the studies in the literature

In this paper, we first present the method of the systematic review and the basic characteristics of the sample we obtained. Then, we present and discuss the results we obtained in these four areas. Finally, we present some recommendations based on them to inform the design of more integral, effective and sustainable real-world interventions based on social norms.

## 2. Materials and Methods

In order to obtain a broader and more balanced sample of empirical studies than the ones that traditional reviews usually rely on, we applied the procedures that are commonly used in systematic literature reviews [73]. Because of our focus on exploring the particularities of intervention strategies and mechanisms in a sample that was as varied as possible, rather than on evaluating their results, we chose to produce a qualitative synthesis [73] of these mechanisms rather than a conventional meta-analysis [74].

### 2.1. Preparation and literature search

The first step after defining the type of empirical literature of interest to the study was defining a detailed protocol following Okoli and Schabram's [73] eight steps to conducting systematic literature reviews (including (1) Purpose, (2) Protocol and training, (3) Literature search, (4) Practical screen, (5) Quality appraisal, (6) Data extraction, (7) Synthesis of studies and (8) Writing the review). Data extraction formats were also designed for relevant activities, and researchers discussed extensively, pre-tested (sometimes several times), and trained on the general protocol and on the use of the specific formats.

The literature search was conducted on the 11th of May 2017 in six widely popular academic databases: Cochrane, Medline, PsycINFO, PubMed, Scopus and Web of Science. We used a Boolean formula to search for title only (in order to keep the number of hits manageable, otherwise initial hits would have been on the tenths of thousands). Keywords are presented in Box 1.

**Box 1.** Keywords used for systematic search

| |
|---|
| "social norm*" OR "descriptive norm*" OR "injunctive norm*" OR "collective norm*" OR "normative*" |
| AND |
| "intervention" OR "field study" OR "field experiment" OR "randomized controlled trial" OR "randomized controlled trial" OR "program" OR "campaign" OR "initiative" OR "change" |

*2.2. Eligibility Criteria and Additional Searches*

Initial hits from the database search (N = 1581) were screened for duplicates. The abstracts of the remaining articles (N = 523) were assessed under our practical screen—PS criteria [73]. At this stage, we included papers with an abstract in English (but text in any language), that presented empirical studies conducted in field settings, and that explicitly included social norms in their design as well as some type of behavioral outcome measure (in case of doubt, this was assessed from reading the full paper). Through this process, we excluded books, dissertations, research protocols or posters (because of the difficulty of obtaining digital versions and/or extracting the required information in many of them). We also excluded laboratory experiments and interventions that didn't measure behavioral outcomes (but measured only attitudes, behavioral intentions or norm perceptions, for example).

Then, the full text of the remaining articles was reviewed according to our quality appraisal—QA criteria [73]. Because of the focus of this review, these were relatively loose: we only excluded articles that did not have ex-ante and ex-post measurements of behavioral outcomes or a relevant control group (so as to assess general findings), or that did not describe in detail the methodological and intervention procedures used.

To complement our sample, we used the remaining articles to perform manual backwards searches (studies referenced by the selected articles) and forward searches (studies that cite the selected articles). We also contacted experts in the field and the authors of these remaining articles to ask for published and unpublished studies that matched our initial eligibility criteria. All the additional studies identified were reviewed through the same PS and QA procedures as the rest.

For both the PS and QA reviews, detailed definitions of the criteria were produced. Specifically, PS criteria included references that (i) were published as papers (books, dissertations, posters and research protocols were excluded), (ii) presented studies conducted in field settings (excluding laboratory or conceptual work), (iii) presented interventions that explicitly mentioned using social norms in their intervention design, and (iv) included at least an English abstract (but text in any languages). QA criteria excluded articles with studies that (i) lacked measurement of behavioral outcomes (even if self-reported) or (ii) lacked a clear description of methodological procedures and intervention mechanisms used.

Pre-tests were conducted (three in the PS and one in the QA) and inter-rater reliability measured (with Cohen's $\kappa$ = 0.856 for final test of the PS and $\kappa$ = 0.841 for the QA). The coding process for exclusion and inclusion of studies can therefore be considered reliable [75]. The process from identification to inclusion is summarized in the PRISMA diagram included in Figure 1 [76].

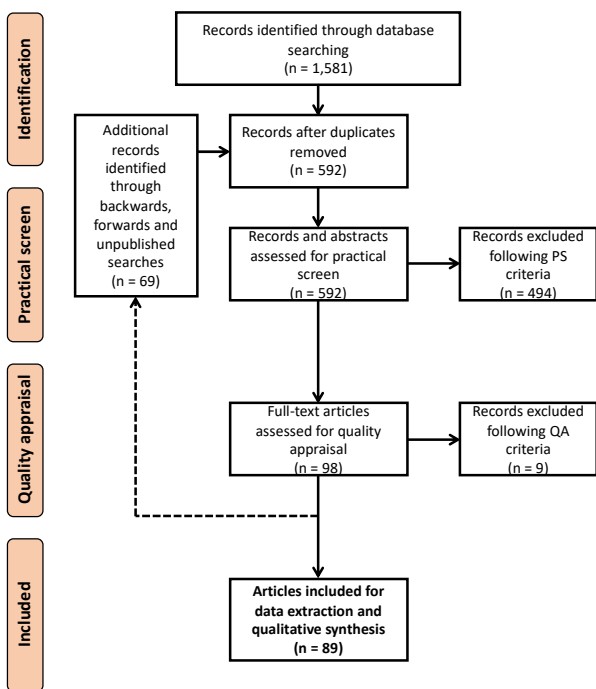

**Figure 1.** PRISMA diagram for systematic review.

*2.3. Data Extraction and Analysis*

Basic information about the study and the intervention in question was then extracted from the papers (such as the country, outcome measures, sample size and characteristics, treatment conditions or target behavior). Additionally, researchers coded the use of the two main dimensions described above (situated, remote, groups summary information and exposure to behavior and opinions), which was also previously pre-tested measuring interrater reliability (Cohen's $\kappa$ = 0.733–1.000 for the two dimensions, thus also considered adequate [75]). Finally, we also extracted qualitative information about the intervention mechanisms that were used in each article and conducted a thematic analysis to find relevant categories that fitted the three types of local determinants of behavior (specifically, we copied and pasted the descriptions of intervention procedures that were used, such as workshops, emails or distribution of flyers, and then grouped these into the mechanisms presented below). For further reference, the complete list of studies selected can be found in Appendix 1 and the full coded set can be accessed in the Supplementary Materials (Table S1).

Additionally, in order to expand our analysis and explore in a more systematic way the interrelations between these elements, we applied a Multiple Correspondence Analysis–MCA [77] on the coded categories to explore the pattern of relationships between the main dimensions, the mechanisms, and the basic characteristics of each study. MCA creates two or three dimensional maps of components according to their variance [77], which allows to identify the categories that tend to appear together in a specific sample (in our case, the dimensions, mechanisms and characteristics that tend to be used together).

## 3. Results

After following the procedures above, our sample consisted of 89 published and unpublished articles that contained a total of 92 studies reporting the results of behavioral change interventions based on social norms in field settings. As already discussed, the studies are journal articles or unpublished manuscripts that explicitly use a social norm approach and that measure some type of behavioral outcome.

The studies in our sample were published between 1991 and 2017. Around two-thirds (66%) of them were published from 2010 onwards, and around half (47%) between 2013 and 2017. Five articles were yet to be published. Studies were conducted in 17 different countries, most in North America (68%), followed by Europe (19%), while only a few were conducted in other continents. Over two-thirds of the studies (66%) were conducted in the US, and only one study took place in multiple countries. All but three of the articles conducted a single study, which spanned 12 different topics. Over half of the studies (52%) targeted alcohol consumption. This reflects which issues are targeted by behavioral interventions and where, as published in the literature. These biases are interesting to note, and while our study is to some extent representative of the published research, it suffers from this limitation. Figure 2 below presents an overview of the distribution of some of these characteristics, while Table 1 below presents the frequencies of the categories that compose them.

Regarding study designs, the sample sizes varied greatly—from as low as 21 to almost 400,000. These were grouped into four categories which yielded fairly equal distributions: less than 200 subjects, 201 to 500 subjects, 501 to 1000 subjects, and over 1000 subjects. Around two-thirds of studies (63%) identified their participants as students, which is a classic limitation [78,79], but there was a wide variety of participants in the rest of the studies (see Table 1). Most studies (around four-fifths), used different treatment conditions and a control group as design, while the rest used pre/post designs without conditions or control groups.

Regarding data collection methods, the vast majority of studies used surveys and questionnaires. Only four studies used qualitative methods, while the rest used quantitative. Regarding outcome measures related to behavior, over two-thirds of the studies (70%) used self-report measurements only, while less than one-third (30%) used some kind of "objective" recording of behavior (either traces or direct). Additionally, around half of the studies used a wide variety of non-behavioral outcome measures, the most common of which are normative perceptions, risks and consequences of behavior, motivation, intentions or self-efficacy, and intervention exposure or relevance.

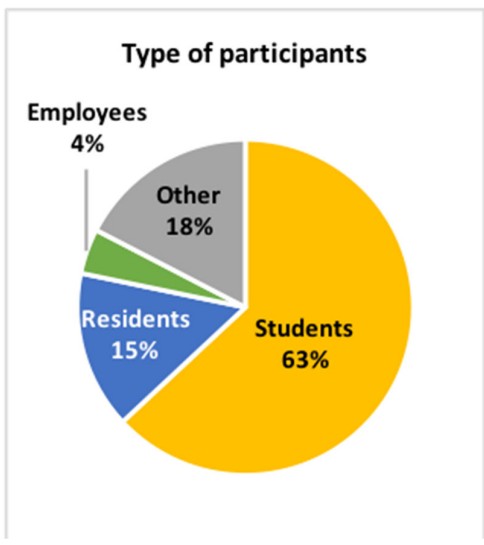

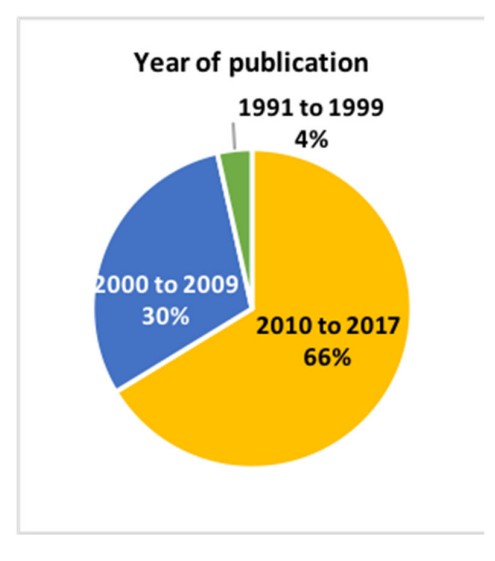

(a) (b)

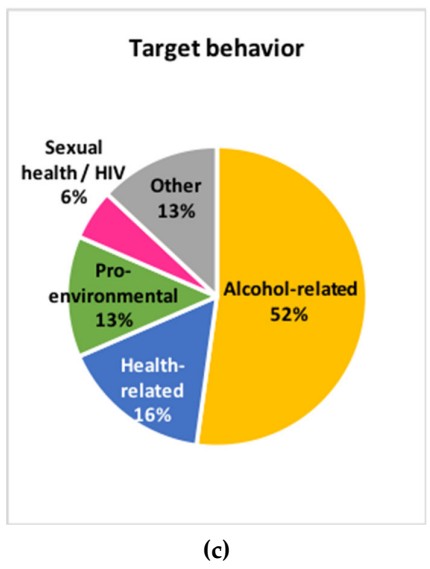
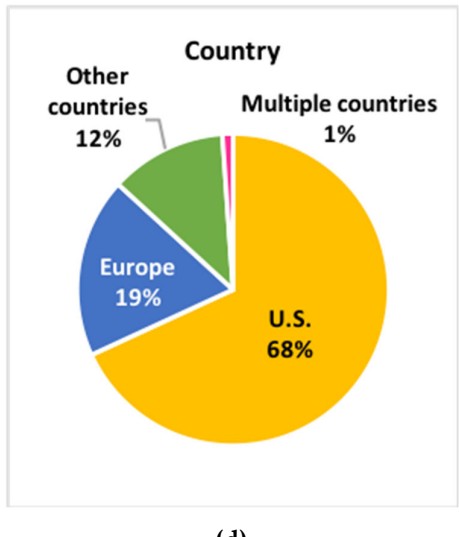

**Figure 2.** Distributions of type of participants **(a)**; year of publication **(b)**; target behavior **(c)** and country of participant in the study sample **(d)**.

**Table 1.** Frequencies of countries, target behaviors and type of participants in study sample.

| Countries | | Target Behaviors | | Type of Participants | |
|---|---|---|---|---|---|
| | Frequency | | Frequency | | Frequency |
| US | 63 | Alcohol consumption | 48 | Students | 58 |
| UK | 7 | Health behaviors | 14 | Residents | 14 |
| Netherlands | 3 | Pro-environmental * | 13 | Employees | 4 |
| Australia | 2 | Sexual health / HIV | 5 | Drinkers-drug users | 4 |
| Canada | 2 | Bullying / harassment behavior | 4 | Passers-by | 3 |
| Colombia | 2 | Gambling | 2 | Shoppers | 2 |
| Germany | 2 | Tax and retirement | 2 | Users of public restrooms | 2 |
| Switzerland | 2 | Counselling service use | 1 | Cyclists | 1 |
| Austria | 1 | Ethical behavior | 1 | Doctors | 1 |
| Denmark | 1 | Rating of interviews | 1 | Patients | 1 |
| Ecuador | 1 | Use of pedestrian crossings | 1 | Pension fund participants | 1 |
| Iran | 1 | | | Taxpayers | 1 |
| Kenya | 1 | | | | |
| Multiple countries | 1 | | | | |
| Poland | 1 | - | | - | |
| Tanzania | 1 | | | | |
| Zambia | 1 | | | | |

\* Pro-environmental behaviors targeted included: water and energy consumption, use of sustainable transportation methods for commuting (bicycles), recycling and littering.

Finally, regarding effects, three quarters of the studies (75%) reported statistically significant effects (of $p \leq 0.05$) on the target behavior linked to social norms. Of those that included measures of actual behaviors (as opposed to self-reported measures), 89% reported statistically significant results with small effect sizes (in the few studies that report it) of partial $\eta2 = 0.01$–$0.04$ (see for example Celio et al. [41], Brudermann et al. [81] or Wally et al. [40,80,81], which is consistent with previous reviews such as John et al. [11]). Most studies (71%) measured effects until six months or less after the intervention was measured, with around one third of the total (33%) doing it only one month or less. Only around one-quarter of the studies (27%) measured these effects for more than 6 months after the intervention.

### 3.1. Situated and Remote Interventions

A first fundamental distinction in the way that researchers apply behavioral change interventions based on social norms is the one between those interventions that are situated, applied at the point of delivery of behavior (in situ), and those that are applied remotely, in a different context from the one in which the target behavior happens. As Suchman [82] noted, action is situated, and so is cognition [83], therefore it is likely that interventions situated where the action take place might have a different impact from remote ones. To use a typical example in our sample, it's not the same to distribute fliers with normative information in the context and moment in which the target behavior is happening (for example through coasters and table mats in bars to tackle drinking behaviors—see Moore et al. [84]), compared to distributing the same information remotely, one week before in an unrelated context (for example through an email or a postcard mailed at home—see Schultz et al. [15]).

Social norm interventions that are applied by researchers on the context where the target behavior happens (situated) rely on the idea that immediate contexts have some influence over behavior. Consequently, modifying in some way the context (which includes the physical environment but also the information that is available to people and the interactions that happen on it) should lead to behavioral changes. As already mentioned, researchers in psychology have long explored how the effect of social norms on behavior is not uniform over different contexts or times. In their famous "Focus Theory of Normative Conduct", Cialdini, Reno and Kallgren [42] argue that "norms should motivate behavior primarily when they are activated (i.e. made salient or otherwise focused on)" [42]. Similarly, Aarts and Dijksterhuis [44] explore the concept of situational norms, showing that certain environments can automatically activate normative behavior, especially when (i) goals to visit the environment are active and (ii) strong associations between environment and normative behavior are established. Interventions that change local contexts have been shown to have important effects on transforming behavior. Among many examples, Lahlou and colleagues [85] showed how in-situ interventions to increase the water intake of children at home can achieve increases of more than 700% in the target behavior by leveraging the social (via a social support forum), physical (by providing water bottles) and psychological (providing training) determinants of behavior.

In our sample, less than half of the studies (39–42%) applied interventions on the context where the target behavior happens. Of these, around half (21–54%) were applied only in situ, while the other half (18–46%) were combined with remote interventions (away from the context were the behavior happens). Most interventions in this group were conducted in the U.S. (24–61%), targeted alcohol-related (10–26%), pro-environmental (11–28%), or health-related behaviors (8–20%) behaviors, and were applied on students (15–38%) and residents (10–26%). Most interventions that were applied in-situ did so by introducing specific bits of normative information messages in marketing materials such as fliers, cards, letters, posters and signs. Some interventions also used a wider range of materials, including t-shirts, water bottles, coasters, and stickers, and others also included media and digital platforms such as TV, radio, magazine, and theatre ads, as well as email messages, websites, and even interactive software.

These interventions explicitly or implicitly followed the strategies of the Social Norms Marketing —SNM approach [86,87], which is "a means of correcting norm misperceptions that involves publicizing (marketing) the actual rate of the misperceived behavior via the media, posters, emails, etc." [23] (p. 341). For example, Schultz and colleagues [15,18] used door hangers and letters delivered at home to try to increase recycling and reduce water consumption in households, Lapinski and colleagues [24] used posters in toilets to increase handwashing behaviors, and Payne and colleagues [22] used signs in shopping carts to try to increase vegetable and fruit expenditure. In our sample, this information was often complemented with non-normative pieces of relevant information, such as the risks or benefits of target behaviors, its related rules, regulations, and existing initiatives, and tips and goals to direct action.

Apart from providing information about actual rates of behavior (and the opinions around it), interventions have also attempted to communicate that a behavior is or isn't typical or desirable using

marketing materials that are allusive to social vigilance or to broader campaigns. For example, Bateson and colleagues [33] and Brudermann and colleagues [80] used posters with images of eyes to influence littering and payment behaviors, and Paluck and Shepherd [35] used wristbands to express support for a broader social norms campaign against harassment.

Rather than relying only in these normative effects, some researchers also complement normative information with other mechanisms that support behaviors and interactions in the context where behaviors happen. In our sample, one way they do this is by supplying objects that support a target behavior (if the goal is to increase its occurrence). For example, Flüchter and colleagues [17] provided bikes to try to increase cycling to work, and Lahlou and colleagues [85] provided water bottles to increase water intake by children. In other cases, researchers also created interaction and social control processes with their interventions, including digital forums to discuss normative information and provide mutual support [85], mime-artists that made fun of traffic offenders and praised norm-compliers [6], or thumbs-up and thumbs-down cards for people to express approval or rejection of observed behaviors on the street [6].

Of course, some of these mechanisms to support behaviors, interactions, or control aren't linked to social regulation processes per se. Nevertheless, as happens in the studies of our sample, they can transmit powerful normative messages as well, or make the norm salient, and as such they can have powerful effects on social norm perceptions and more broadly in social regulation processes. Specifically, they actively support social control, they make visible the preferences and opinions of group members, and they communicate that someone is willing and able to act in favor or against a change initiative.

Other researchers that attempt to leverage social norms to change behavior in the real world do not apply their interventions in the context where the behavior happens, but rather in different and often un-related contexts. These interventions are based on the idea that, if people's behavior is influenced by their "subjective perceptions of the norm" ([3], p. 181) which "become a reality and a guide for their own behavior, even when the perceptions are inaccurate" ([3], p. 183), then changing those perceptions could change behavior in an enduring manner. In practice, this means that providing people with certain pieces of relevant normative information can ultimately persuade them of acting differently in the future (whenever they encounter the context where that norm and that behavior is relevant). Researchers have extensively used successful remote interventions to change behaviors, for example by providing normative information via a web-based survey that is related to decreases in overall alcohol intake [37,88]. While the connection to relevant behaviors and situations might sometimes be more difficult to achieve through these interventions, their advantage is that participants are usually much more focused on the intervention mechanisms than when performing the target behavior at the same time. Another advantage of this approach is practical: it is easier to give a message to subjects where one can easily gather them, than to install the message in all the sites where the behavior may occur. That is especially the case if, for example, those making the intervention are academics and their target is students.

As a matter of fact, most interventions in our sample (66–72%) applied remote interventions, with the majority of them (53–80%) not including any direct action on the target context. This means that while around half of situated interventions included some remote action as well, only a fifth of remote interventions include some situated initiative. As for their basic characteristics, most remote interventions were also conducted in the U.S. (48–73%), targeted alcohol-related behaviors (48–73%) and were applied on students (52–79%). Although they show similar characteristics to situated interventions in this respect, most remote studies were concentrated in these three categories (compared to situated interventions, which were distributed more evenly).

Here also, the most common way in which remote interventions were applied is by providing participants with normative information messages. Along with widely used fliers, letters, ads, and other marketing materials including very similar messages, a large number of studies used email communications and web-based surveys to apply a Personalized Normative Feedback—PNF [40,89] approach. PNF is a popular intervention method that attempts to correct "norm misperceptions" by "collecting participants' self-reported incidence of some behavior and their perception of the

incidence of this behavior among their peers and then providing them with the actual incidence of the behavior" [23]. In our sample, it was mostly used in interventions focused on alcohol consumption among students, where participants received an email, completed a survey, and then received feedback about their norm perceptions and about their consumption levels (which were then compared to the ones of a reference group).

And just as with situated interventions, remote initiatives attempted to communicate normative information in ways that are different from the usual group's rates of behavior and opinion. For example, some authors used face-to-face interaction of experimenters with participants to discuss normative information [19,90,91], while others used videos with real-life stories [92]. Also, in a similar way to situated interventions, studies very often complemented this with the same type of non-normative relevant information (e.g. why this behavior is "better" than the alternatives).

In addition to remote interventions transmitting information to participants, some studies also generated discussions between participants about normative information they had previously provided. This was also complemented with discussions about other relevant information and personal experiences. For example, discussion topics led to participants' "methods to keep themselves safe in party environments" ([93], p. 6), "accurate information about the prevention of HIV transmission" ([94], p. 443), or "explanations for men's perceptions of false accusations of assault" ([36], p. 724).

And finally, some remote interventions in our sample were also complemented with mechanisms designed to support or facilitate target behaviors through the embodied competences [27] and motivations of participants. In this respect, the most popular one included training and skills building sessions that were relevant to the target behavior. This was sometimes done with participants, but also with key actors or referents that might influence the rest of the group. For example, Mogro-Wilson et al. [92] taught drug refusal skills to participants, Lahlou et al. [85] conducted online coaching sessions on water intake benefits, and Paluck and Shepherd [35] trained social referents (defined as "highly connected and chronically salient individuals in a community" [35], p. 899) to write and perform drama skits about common types of harassment in a school assembly.

There are also examples of interventions that generated commitments to action in participants or imposed relevant financial incentives and penalties. For example, this included students signing a contract with strategies to reduce smoking [95], monthly letters to hospital employees giving small incentives for buying healthy food at the cafeteria [21] and imposing fines to households that consumed significantly more water than average relative to their occupants [6]. Finally, one intervention aimed at reducing alcohol and drug consumption among adolescents also embedded normative information in psychological therapy sessions for participants [96].

In summary, the first dimension opposes what seems to be a majority approach of attempting to transform internal perceptions away of the contexts where the target behavior happens (remote), to a less popular approach of modifying the environments in which those behaviors actually happen (situated). Despite their differences, the vast majority of both types of interventions attempted to transform behavior by either distributing pieces of normative information messages, or by creating interaction processes between participants. We now turn to exploring this second relevant dimension in the way in which interventions are applied.

*3.2. Interventions Based on Group Summary Information and on Exposure to Behaviors and Opinions*

The second fundamental distinction we found in the way that researchers apply these interventions has to do with the way in which they attempt to communicate normative information (the typical or desirable norms of behavior that are meant to produce the change). On the one hand, most interventions chose to give participants a persuasive message about the behavior or the opinions of the group (group summary information) [3], in some statistical format such as the "Four out of five college students wash their hands EVERY time they use the bathroom" in Lapinski, Maloney, Braz, and Shulman ([24], p. 27). On the other, many interventions also attempt to transmit this same information by creating the conditions for participants to be exposed to other people's behaviors or

expressed opinions. This includes, for example, generating collective discussions between participants in the spirit of Lewin's [97] seminal work [36,98], representing drama skits with personal experiences or simulated situations [35], having social referents or well-known figures publicly endorsing or rejecting target behaviors [35], and even creating public demonstrations [6]. In a way, this dimension is about "cold" (anonymous information only) vs "hot" (involving exposure to behaviors and opinions) communication of the norm.

A similar distinction is defined by Tankard and Paluck [3] as the difference between normative information that is communicated through "summary information about a group" [3], and that which is acquired by perceiving group member's "behavior or expressed opinions" ([3], p. 185). Researchers have explored extensively how receiving summary information about the behaviors and opinions of a group can shape the behavior of individuals. Reading simple messages about "how many people", "how often" and "how positively my group feels" [3] (p. 189) about specific behaviors has been shown to have important effects on individual normative perceptions and behaviors, as many landmark social norm experiments have shown [18,46,49,50]. These "argument-based messages" ([99], p. 113) are cognitively-oriented modes of influence [99,100] close to paradigmatic modes of thinking [101].

Although they are popular in the general psychological literature about persuasion, argument-based messages are also "at odds with lived experience" ([99], p. 113). In the case of social influence and normative processes, most of the information that shapes our normative perceptions in everyday life comes from our interactions with others and from paying attention to their behaviors and opinions. Researchers in this respect have focused on how there are certain individuals that are especially influential over our perceptions of norms [35]. These social referents, as they call them, are "highly connected and chronically salient actors in a group" ([35], p. 899) that are weighted more heavily in their behavior and opinions when people "form their impressions of the norms of their reference group" ([3], p. 185). These referents can be real group members, but they can also be imaginary or role-models as well. Researchers have applied effective interventions based on these ideas, some even successfully changing the behaviors of a whole group by targeting social referents only [35]. This is supported by Kareeev's [102] research on how limitations of working memory mean that people usually rely on small samples of $7 \pm 2$ elements to make inferences about whole populations. In our case, for example, this means that when a person sees a group member around her recycling or expressing favorable opinions about it, she will infer that "recycling is typical and desirable for her group" ([3], p. 184).

As happens with remote interventions, group summary information interventions also seem to have a practical advantage: it is easier to design interventions that deliver information only, rather than setting up interaction contexts where the subjects are directly exposed to influence or persuasion by other group members. But while many studies assume that the two types of interventions are equivalent, they are based on different conceptions of social influence and persuasion processes. Moreover, they can often convey very different normative information, as statistics that describe average or majority trends in a population can differ greatly from the effects of confronting actual individuals that will likely hold different positions, as Michaeli and Spiro´s [103] research points out (As Reviewer 2 rightly argued, for example, "if half the population ranks the acceptability of abortions on a 1 to 5 scale as '1' while the other half ranks it as '5', the group-summary statistic that reports an average acceptability of 3 could have a totally different effect on behavior compared to exposure to individuals from both sides of the debate, which could lead, in some cases, to the emergence of biased norms").

Interventions that are applied using group summary information rely on the idea that people's perceptions of the norm can be transformed by providing them with information related to the rates in which group members engage, approve or disapprove a specific behavior. Maybe because of how simple it can be to produce and transmit these messages, and because it is "in some ways the most straightforward manipulation of a perceived norm" ([3], p. 189), this type of approach is by far the most widely used in our sample when applying social-norm interventions (82 studies—89%).

As the studies in our sample show, interventions based on group summary information tend to be somewhat effective in transforming a very wide range of behavior in real-world contexts. Nevertheless, around one-fifth (17–19%) of the studies in our sample (most of which used group summary information) did not find effects on behavior linked to social norms. In the broader psychological literature, there is evidence supporting the idea that interventions based only on giving normative information can be ineffective to change behavior [104–107], and there is even evidence that message content or quantity of information can be unrelated to the effectiveness of social norm interventions [24]. On the other hand, combining normative information with other change tools (such as awareness of the problem, tools for action and prevention and a written promise- see Elias-Lambert and Black [108]), as well as making people conscious about the frequency in which they engage in a behavior and providing feedback about it [109,110] have both proven to be effective. Indeed, the vast majority of interventions in our sample mixed normative information with other mechanisms such as discussions, factual or context information, or calls for action and proposed goals.

In our sample, most studies (82–89%) relied in one way or another on delivering "argument-based" [99] messages with group summary information. This amounts to persuasion (using the logical, argumentative route) rather than influence. Most of those interventions (61–66%) used only this type of information without deliberately exposing participants to the behaviors or opinions of others. Because of this distribution, the profiles of the interventions that use summary information are very similar to those of our whole sample.

The two most common ways in which these interventions were applied in our sample was either by disseminating group summary information to the target participants as a whole (a strategy close to the Social Norms Marketing approach) or by adding to this information some feedback about how the participant´s individual behavior or perceptions compare to them (a strategy close to Personalized Normative Feedback). As was reviewed above, this included both situated and remote interventions. For example, studies that use the former strategy typically use marketing materials (ads, fliers, posters…) to distribute messages like "Most Northwestern Montana's Young Adults (88%) Don't Drink and Drive" ([111], p. 869). On the other hand, studies in the later group use more personalized communication channels (especially emails linked to web-based surveys) to provide graphs and messages like " the number of occasions you drank was 4 times a week… You told us that you believed that the average student drank five times a week … The actual drinking norm for students at the University of Washington is 1.5 times a week … you drink more than 91% of other college students" ([89], p. 436).

Researchers seemed to choose opportunistically the materials that worked better for their specific context and hypothesis: for example, signs were placed in shopping carts to increase produce demand [22], or coasters and glasses were marked with campaign information to reduce drinking [84]. Also, while some researchers relied on a single message and material, for example by only displaying normative feedback after completing a web-based survey through PNF [109], others combined several promotional materials that went from fliers and posters, to coasters, stickers, glasses, and meal planners [84].

Also, when applying these interventions, researchers often manipulate the way in which these messages are presented in order to test different effects on participants' perceptions. For example, some researchers have explored the differences between using descriptive or injunctive norms [98], or how private or visible the target behavior is [24]. And just as with situated and remote interventions, researchers combined these normative messages with other actions that support or enable the target behaviors, such as providing non-normative information, tips for action, or different objects, for example.

In addition to providing generic summary information, some interventions include different resources to allow participants to collect information about behaviors and opinions of others so they can experience social norms through personal interactions [112]. This is the first source of normative information described in Tankard and Paluck's [3] research, and is based on the idea that "individuals' subjective perceptions of norms are not derived directly from a comprehensive survey

or a census" [3], but rather from "their unique and local experience" [3]. Instead of just providing a message with rates of behaviors or opinions, these interventions are based on creating interactions between participants which make visible the way group members are acting, their opinions on a topic, or their willingness and efforts to change (which arguably is how most normative information is transmitted and enforced in everyday life—see Paluck [112] and also the "vigilante effect" in Lahlou [27] p. 126). Interventions based on making visible certain group behaviors and opinions have obtained important results in city-wide scales, for example managing in a 7-million-people city to reduce indicators like homicides by 70% [113], deaths in traffic accidents by 65% [114], and per capita water consumption by 46% [115] in less than a decade (some of them included under Mockus [6]).

Around one-third (29–31%) of the interventions in our sample were based on direct exposure to the behaviors and opinions of others. Most of those (21–72%) did this in addition to generic summary information, were conducted in the U.S. (20–69%), targeted alcohol-related behaviors (14–48%), and were applied on students (17–59%).

In our sample, the type of actions used by interventions in this group included collective discussions between participants, face-to-face interactions and workshops between researchers and participants, and online support forums, but also theatre skits, videos, law enforcement, selling of bracelets by social referents, cartoons, and even public demonstrations. For example, Balvig and Holmberg [95] used discussions between pupils of a school about their own normative "misperceptions regarding cigarette smoking among their peers" [95], McCoy and colleagues [39] used a publicly displayed poster in a clinic for patients to paste a sticker when they attended three consecutive antiretroviral therapy visits, and Agha and Van Rossem [94], Mogro-Wilson et al. [92], and Paluck and Shepherd [35] used videos or theatre skits portraying relevant personal experiences and real-world situations.

These varied actions made visible three main types of information: group member's opinions, group member's behavior, and group member's willingness and efforts to achieve a certain change. For example, interventions based on collective discussions and forums between participants make visible their approval or disapproval of certain behaviors (desirable behavior), and in some cases also their reported behaviors on the matter (typical behavior). In a similar manner, theatre skits, videos, and other materials that make visible participants' behavior, their personal experiences, or simulated stories (such as cartoons–see Lapinski [24]), all exemplify behavior. Finally, actions such as law enforcement, social referents endorsement of the change interventions (through using and selling campaign bracelets, for example—see Paluck and Shepherd [35]), and public demonstrations, all show that people in the group are willing and actively trying to change the target behavior.

In summary, this second dimension describes two different ways in which normative information, which describes the typical and desirable behaviors in a group, is communicated to achieve behavioral change. On the one hand, the most popular approach is based on providing summary messages through a wide variety of marketing materials (group summary information), while, on the other, a less popular approach relies on creating direct interactions between participants that make visible their behaviors or opinions (exposure to behaviors and opinions).

*3.3. How are These Dimensions Combined in Interventions?*

Of course, the two dimensions described above are not applied independently, but are rather combined in very different ways in order to achieve their desired effects. Some studies used both remote and situated interventions, and others used both group summary information and exposure to behaviors and opinions. By representing these dimensions in a simple Cartesian system presented in Figure 3, we propose an analytic framework that can help understand and characterize how normative information is transmitted in behavioral change interventions based on social norms.

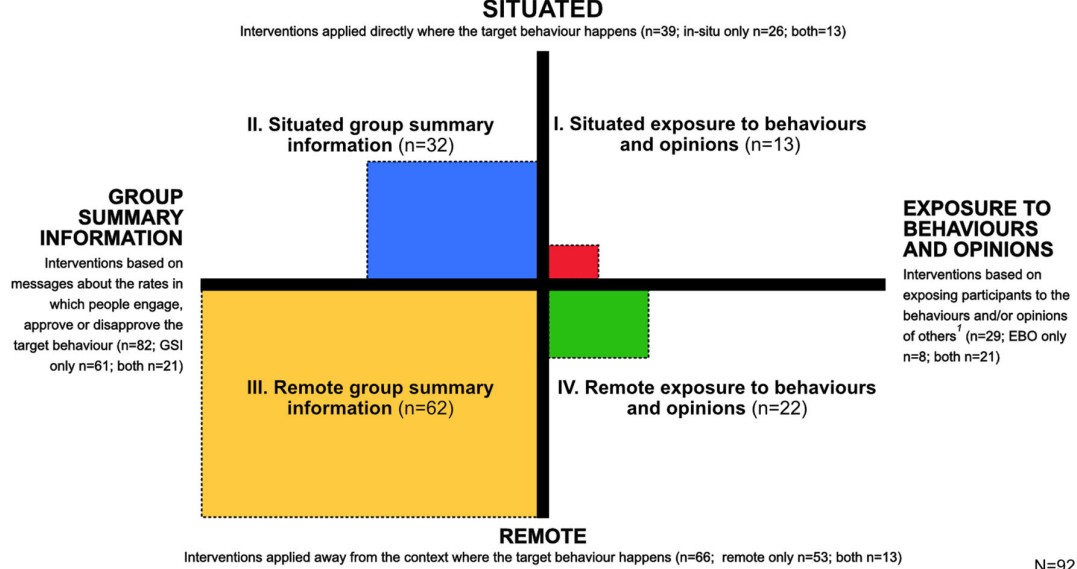

**Figure 3.** Intervention dimensions and popular applications.

As displayed in the diagram, where the squares are proportional to the number of sampled interventions in that quadrant, interventions that use group summary information are much more popular in the literature, especially those that do it away from the context in which the target behaviors are performed. This distribution is not surprising, considering the lower difficulty of creating normative messages compared to creating more complicated situations, and of broadcasting it where participants already are or can easily be assembled instead of the contexts where the behavior happens. This first result suggests that there is a practical bias in intervention design: easier interventions are more popular.

Exposure to behaviors and opinions, which arguably is how we collect most normative information in our everyday social interactions [3,112], has been much less explored in academic research. More than half of studies (54–57%) used remote interventions only, more than two-thirds (67–71%) used only group summary information. Very few interventions (11–12%) were applied in-situ and using exposure to behaviors and opinions, while even less (5–5%) opted to test all four quadrants of our framework.

In addition to that, the studies in these four quadrants used a wide range of intervention mechanisms that leveraged social, psychological and physical determinants of behavior to achieve their ends. In our sample, we found 16 mechanisms that studies used to achieve the desired effects, and that complement the ones described by authors like Miller and Prentice [23], Paluck [34] and Bicchieri [2]. These are presented in Table 2 according to which layer of installation theory they target.

**Table 2.** Intervention mechanisms.

| Layer | Mechanisms (application in sample) | Common applications in sample |
|---|---|---|
| **Social** | Transmitting group summary information messages (situated–remote) | Messages summarizing normative information about a group, for example after completing survey or through email, or in promotional materials such as fliers, letters posters, signs, stickers and adds |
| | Exposure to behaviour and opinions (situated–remote) | Face-to-face interaction, videos, staring eyes in walls, drama skits, public rejections by social referents and public figures, as well as public demonstrations |
| | Generating discussions about normative information (situated) | Sessions with face-to-face discussions among participants |
| | Law and policy enforcement (situated) | Direct enforcement and control in specific contexts |

| Layer | Mechanisms (application in sample) | Common applications in sample |
|---|---|---|
| | Mutual regulation (situated) | Creating situations or distributing objects that promote mutual regulation among participants, such as mime-artists, cards or whistles |
| | Social support (situated) | Creating digital forums to discuss normative information and provide mutual support |
| Psychological | Providing factual/context information (situated–remote) | Information about risks/benefits of engaging in target behavior in signs, leaflets and posters, and context information in leaflets, fliers, or through digital media |
| | Providing tips and guides for action / goal setting (situated–remote) | Steps and goals for engaging in target behaviors in leaflets, fliers, or through digital media |
| | Generating discussions among participants (indirect) | Face-to-face sessions to discuss information relevant to target behaviors |
| | Generating training/skills building sessions (remote) | Face-to-face sessions to train and teach skills to participants and relevant referents |
| | Creating commitments to action (remote) | Contracts or symbolic actions to generate behavior-relevant commitments |
| | Providing incentives (remote) | Applying financial incentives or penalties for engaging in target behaviors |
| | Providing psychological therapy (remote) | Regular therapy sessions for participants |
| Physical | Modifying environments (situated) | Fixing promotional materials (such as posters or signs) or providing objects that support target behaviors (such as bikes, water bottles or whistles) |
| | Distributing papers/objects (situated–remote) | Distributing promotional materials (such as fliers or t-shirts) or objects that support target behaviors (such as bikes, water bottles or whistles) |
| | Generating interactions with digital platforms (remote) | Use of digital platforms (such as computers and smart phones) to distribute intervention materials and information |

In order to explore further how these interventions were applied and their characteristics, we conducted a Multiple Correspondence Analysis—MCA to explore what experimental characteristics tend to be combined in our sample. MCA is a technique to analyze the pattern or relationships between different categorical variables which produces the principal components that account for the variance in the data [77]. The first two or three components often account for most of the variation in the data (or inertia), and this produces a map in which proximity denotes that certain modalities of variables tend to occur together.

To conduct our MCA, we included the two dimensions described above (situated/remote/Group Summary Information/Exposure to Behaviors and Opinions), together with the most popular (specifically, those with 10 or more occurrences in the sample) intervention mechanisms in Table 2 and the information on whether the study registered significant effects on behavior linked to social norms. We also included information on the target behaviors, types of participants and year of the study as supplementary (passive) variables. The results show two main components that account for 61.5% and 12.5% of the variance respectively, as the simplified diagram in Figure 4 presents.

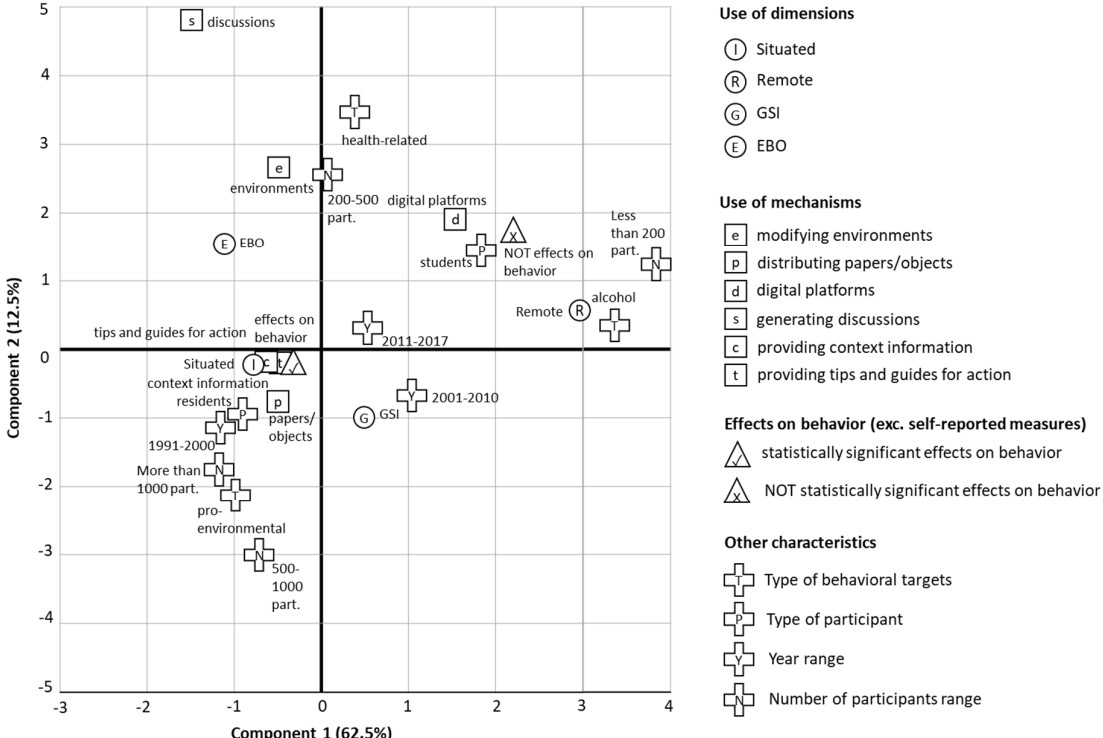

**Figure 4.** Multiple Correspondence Analysis. GSI refers to Groups Summary Information, and EBO to Exposure to Behaviors and Opinions.

Because of simplicity and space considerations, only the general associations that were drawn from the analyses will be reported here at the risk of losing some of their details and nuances. The complete MCA output and coordinate plot can be found in the Supplementary Materials (Figures S1 and S2 respectively). Two general conclusions emerged from this analysis:

1) The main axis, which accounts for most of the variance in the sample, opposes "lighter" interventions applied remotely, using GSI and digital platforms mainly on students (which as we have seen constitutes the bulk of our sample and is associated with quadrant III), with more complex interventions done in-situ, using exposure to behaviors and opinions, and relying on a wider variety of interventions mechanisms (associated with quadrant I). Characteristics associated with each of these groups are presented in Table 3.

**Table 3.** Tendencies of dimensions, mechanisms and other study characteristics.

| Characteristic | Group 1 (component 1 +) | Group 2 (component 1 -) |
|---|---|---|
| **Dimensions** | Remote<br>Group summary intervention | Situated<br>Exposure to behaviors and opinions |
| **Intervention mechanisms** | Digital platforms | Discussions among participants (about normative and non-normative information)<br>Providing context information<br>Providing tips and guides for action<br>Modifying environments<br>Distributing papers and objects |
| **Target behaviors** | Alcohol consumption<br>Health behaviors | Pro-environmental behaviors |
| **Participant types** | Students<br>Employees | Residents |
| **Period of publication** | 2001–2010<br>2011–2017 | 1991–2000 |

| Characteristic | Group 1 (component 1 +) | Group 2 (component 1 -) |
|---|---|---|
| **Number of** | Less than 200 | Between 500 and 1000 |
| **participants** | Between 200 and 500 | More than 1000 |

2)  When including only studies that recorded behavior (excluding self-reported measures) finding relevant effects on behavior appears more frequently among interventions in the group 2 (quadrant I). On the other hand, not finding effects is more frequent among interventions in the group 1 (quadrant III). Nevertheless, these results should be taken with caution, as they are based on a limited subsample of 28 studies (in which 25 found effects and three didn't), and there is a great diversity of experimental contexts, targets and treatments in them.

Due to the low number of studies in this subsample that are linked to quadrants I and III, and because of the configuration of the main axis in the MCA, we hypothesize that these associations might be linked to the opposition between remote and situated interventions, as displayed in Table 4. This of course must be systematically tested based both on previous literature and new experimental data in order to assess its plausibility.

**Table 4.** Effects on remote and situated interventions (excluding self-report).

| | Studies with effects on behavior linked to social norms (excluding self-reported measures) | |
|---|---|---|
| | **Yes** | **No** |
| Remote only | 3 | 2 |
| Situated only | 21 | 1 |
| Both | 1 | 0 |
| **TOTAL** | **25** | **3** |

## 4. Discussion and Recommendations for Policy Application

This review describes some of the main characteristics of a relatively large systematic sample of behavioral change interventions based on social norms applied in real-world settings. Its main focus is to describe the main strategies (dimensions) used to operate those interventions (situated, remote, group summary information and exposure to behaviors and opinions). We also describe the range practical physical, psychological and social intervention mechanisms used, and how they tend to be combined in the literature. Though we draw on some of the mechanisms and strategies that have been described by influential texts in the field [2,23,34], our review significantly expands their range, provide practical demonstrations, and a systematic framework to interpret them. These efforts are important, because they enlarge and shed some light on the range of possibilities that researchers and practitioners have when designing and implementing social norm interventions to change the behavior of people in real-world settings.

The variety of interventions included in the sample speaks about the popularity of the social norm approach to tackle a wide variety of social issues and policy challenges [11], most of which are directed towards increasing the social and environmental sustainability of their target contexts. Although our sample includes studies that address 11 types of target behaviors in 12 types of participants and 16 countries, there are also clear patterns in it. Namely, the majority of the studies focus on healthy behavior (mainly reducing alcohol consumption) followed by pro-environmental behavior, are applied among students in the U.S. from 2010 onwards, and are applied remotely using group summary information messages. Most studiesuse only self-reported measures to assess behavioral changes (surveys where participants report their own behavior), and do so only 6 months or less after the intervention was finished.

But despite their common objective of influencing normative perceptions to test changes in behavior, the studies in our sample used very different intervention strategies. These differences pertained to some of the most essential aspects of interventions, including the context in which they were applied (situated versus remote) and the types of normative information leveraged (group summary information versus personal exposure to behaviors and opinions). And while most studies are not particularly clear and don't give enough details about these dimensions, or about why they

chose one rather than the other (or a combination of them), as we have shown these options can be traced to rather different assumptions about human psychology and behavior (and in practice, very different applications).

On the one hand, dimension 1 represents a very old debate in psychology and in the social sciences in general, which broadly pertains to the degree in which factors that are external (such as the physical and social environment) or internal (such as representations, competences or emotions) to individuals can predict behavior (and can thus be influenced and changed). This sends back to the classic notions of locus of control, and also to the fundamental attribution error (the bias towards attributing someone's behavior to personality characteristics rather than to the context—see Ross ([116], pp. 173–220). As noted above, we know that several types of factors coalesce in situation to produce behavior. Unsurprisingly, interventions included in our sample have focused on different aspects of the three layers of installations to achieve behavioral changes, some leaning more towards changing the material and social environments in which behavior happens, and others relying more on transforming internal perceptions in the hope that this will influence future behavior. In practice, the strategy used in social norm interventions can try to make the norm salient in context (situated) by modifying the environment, or to change the individual representation of the norm remotely, "in general", therefore focusing on embodied competences. Implicitly, this addresses one or the other (external or internal) locus of control.

On the other hand, dimension 2 represents the differentiation between sharing summary information about a group and direct social interaction that exposes participants to the behavior and opinions of others (likely involving more emotions since that is a multimodal experience). Social regulation, and the norms through which it is often expressed, sometimes works through summaries we receive about how groups behave or what are their opinions. But most often, it is also supported and works through everyday social interactions [27,112]: by hearing the opinions of others, by seeing or hearing stories about how they have behaved or would behave in certain situations , and also by learning the efforts they are making to change the behaviors and opinions of others. Direct interaction matters in communication. Where English has only one term, "communication", Russian distinguishes *kommunikatsia* and *obschenie* (общение). *Kommunikatsia* (коммуникация) refers to an exchange of information, a notion that is familiar. *Obschenie*, which has no English equivalent, characterizes a specific field of research in Russian psychology. This term incorporates several meanings of the notion of "communication" as, for example, "human relations", "interaction between individuals", "pooling" and, finally, "sharing" in the religious sense [117]. While the interventions we subsumed under the term "group summary information" are mostly on the level of *kommunikatsia*, the others ("exposure to behaviors and opinions") involve some elements of *obschenie* as the group is made more salient.

In a similar manner to what happens with dimensions, many studies fail to explain why they chose to apply a certain intervention mechanism and not others, and they don't describe in detail which ones and in which ways they were used. Through the 16 mechanisms that we have identified, researchers often influence different physical, psychological and social determinants of behavior in order to support the desired changes. Because of this, in order to advance the understanding of social norm interventions and the possibility of ever creating a reliable framework to design interventions that can effectively inform relevant policy challenges, it is critical to distinguish if, for example, in addition to normative information an intervention gave tips and guides for action rather than general information about the benefits of a change, and if participants accessed this information by reading a formal email or watching a funny video, and why these choices were made. Just as with intervention dimensions, the particular mechanisms used can create very different practical interventions and might even determine their efficacy in particular contexts.

Rather than using a coherent framework to choose and describe interventions, the studies in our sample use a very wide variety of combinations between the different dimensions and mechanisms we have described (to the point that very few interventions use exactly the same configurations in our aspects of interest). Nevertheless, our analysis of correspondence between these elements revealed two main groups. One is much more common, especially from the year 2001 onwards, and

consists on "lighter" interventions that use remote group summary information in digital platforms (and that typically target alcohol consumption and other health-related behaviors among students and employees). The other, less common and more frequently used from 1991 to 2000, uses more complex situated interventions based on exposure to behaviors and opinions through a wider range of mechanisms that include collective discussions, providing context information and tips for action, modifying local environments, and distributing papers and objects (to target mainly pro-environmental and harassment behaviors).

While the opposition between these two groups could maybe be traced to the increasing use of information and communication technologies for research and intervention purposes [118], as well as the increasing popularity of the Personalized Normative Feedback approach to tackle alcohol consumption in U.S. universities [23], they also seem to suggest a preference in the literature towards interventions that are simpler, easier and cheaper to apply, and that are based on information rather than on interaction. But despite the limited size of the group of studies that measured actual behavior (rather than using self-reported measures), we also presented some initial indications that, at least in our sample, the second group of more complex interventions (situated/EBO) could be more likely to produce relevant effects on behavior than the lighter ones (remote/GSI). Although this cannot be interpreted as an indication that situated/EBO interventions should always be chosen to tackle policy initiatives, we do believe that this expresses a need to a more serious consideration in both research and policy initiatives of this type of interventions.

Unfortunately for this endeavor, as we already reported, over two-thirds of the studies in our sample used self-reported measures of behavior only to assess their efficacy (in most cases, the answers of participants asking for the "frequency in which you engage" in the target behavior, collected in surveys applied before and after the intervention). But while this might be a good measure of the intervention´s success in changing the perceived social desirability of the target behavior, or how salient it is for them when recalling their activities, there is no guarantee that actual behaviors were changed. Because of this, we also believe that a more intensive application of remote/GSI interventions that use measurements of actual behaviors is needed to assess their real efficacy.

*4.1. How to Design more Integral and Sustainable Interventions based on Social Norms: "Get Closer"*

Most policy challenges are behavioral or have a strong behavioral dimension. Because of that, the success of policy interventions depends on achieving sustainable changes in the way people act. Behavioral change interventions based on social norms are an increasingly popular way to achieve this. Nevertheless, as we have presented in this paper, there is a wide variety of different types of interventions that can be applied to this end, and there are also many examples of interventions that were unsuccessful to transform behavior (see for example in our sample Lewis et al. [119], Moore et al. [84], or Werch et al. [120]).

Despite the fact that many researchers limit themselves to exploring the conceptual and research dimensions of behavioral change interventions, there are also some that have focused on how the practical design and implementation processes of social norm interventions can be improved. These researchers have made important arguments such as the usefulness of using marketing segmentation of participants [55,121] and of the involvement of participants and stakeholders in intervention procedures [27,122,123]. In our sample, Spijkerman and colleagues [124] have shown how social norm interventions can be effective only among certain populations but not others (specifically, their intervention to reduce alcohol consumption reduced drinking only among binge drinkers, but not among their general sample).

The issues of how it is necessary to check if there is some consensus (declared or implicit) for the need to change before applying an intervention [57], how the research methods applied before interventions affect behavior [125], how the effects of individual level interventions or those that are applied on a single occasion quickly decay [68,126], and the importance of messages being credible and relevant [23], have also been discussed. In our sample, a failed alcohol intervention by Thombs

et al. [127] led to the realization that most participants didn't find the campaign messages credible and didn't understand the intended purpose of the campaign.

By providing an overview of how these interventions have been applied, we do not mean to imply that what has been done in the literature should be blindly copied, or even that it is the best way to go. Rather, by providing a broad view of the mechanisms that have been used in these types of interventions, we hope to enlarge and shed some light on the range of possibilities that researchers and practitioners have when implementing interventions.

In this sense, although testing which of the four quadrants described above are more effective is difficult, real-world policy initiatives should not limit themselves to the remote–information–type interventions that seem to be so popular in the literature. Although usually easier to apply and research, these interventions often fail to take into account and leverage the rich variety of ways in which social regulation can influence behavior. In real world contexts, social norms are transmitted and enforced in much more varied ways than through group summary messages. Rather, social norms are supported through a wide variety of interaction processes: by seeing others behave in certain ways, by hearing their opinions and stories, by receiving instructions or feedback about the "correct" way to behave, by having physical artefacts that support or constraint our behavior or remind us of certain norms, and also by using the competencies and internalized norms we have acquired.

We argue that more effective and sustainable interventions can be achieved by combining the advantages of the different quadrants, mechanisms and applications we have described in this text. While remote interventions often allow a greater access and attention of participants, situated interventions modify the contexts where behaviors actually happen and allow participants to enact the new behaviors. And while group summary information allows correcting normative misperceptions and gives clear normative trends, exposure to behavior and opinions allows demonstration and interaction with social actors. Table 5 below provides a summary of some of the main dimensions and mechanisms that should be taken into account when designing a social norm intervention (for a detailed guide on how to prepare, implement and evaluate a social norm intervention for an applied policy challenge (improvement of domestic work conditions), see Yamin and Hobden [128]).

**Table 5.** Dimensions and mechanisms to create behavioral change interventions based on social norms.

| Situated Interventions | |
| --- | --- |
| Group summary information (Q. II) | Exposure to behaviors and opinions (Q. I) |
| Create marketing material with group summary information to be distributed in the same context where the target behavior happens [24]<br><br>Include credible and strategic messages with the rates of prevalence and support that the target behavior (or related ones) have in a certain population (i.e., if you want to reduce drinking rates among students, show how most of them drink less, or that more disapproved heavy drinking, than usually thought [129])<br><br>Choose strategically the marketing materials that are more likely to be seen and remembered by the highest possible number of participants (i.e., posters, fliers, signs, stickers, adds, etc. [111,130,131]) | Create actions and/or provide objects in the same context in which the behavior happens that make visible (see [35,85,111]):<br><br>▪ Other's opinions (i.e., online forums, face-to-face interactions to discuss opinions)<br>▪ Other's behavior (i.e., theatre skits and videos with personal experiences)<br>▪ Other's change initiatives (i.e., public demonstrations and norm enforcement) |
| Remote Interventions | |
| Group summary information (Q. III) | Exposure to behaviors and opinions (Q. IV) |
| Create marketing materials with group summary information to be distributed away from the context where the behavior happens (see guidelines in quadrant I)<br><br>Benefiting from easier access and targetability, consider: | Create actions and/or provide objects away from the context in which the behavior happens to make visible [90,92]:<br><br>▪ Other's opinions (i.e., collective discussions and forums to discuss opinions)<br>▪ Other's behavior (i.e., theatre skits and videos with personal experiences) |

- Providing personalized feedback to the participants about their own behavior and opinions, and how they compare to the ones from the reference group [132]
- Creating interaction or reflection processes around the normative information shared (such as discussions, forums and other activities [98,133])

- Other's change initiatives (i.e., public endorsements and campaign information)

| Other Support Mechanisms |
|---|

Depending on the intervention contexts and target behaviors, as well as the available resources, consider complementing the intervention with mechanisms such as:

Providing objects or modifying physical environments in ways that make the target behavior easier or more likely (such as bikes or water bottles [17,85])

Arranging external law and policy enforcement of the target behavior, or creating situations and providing materials that allow participants to regulate each other [6,134]; this can be enforced by policing personnel or by fellow citizens through the "vigilante effect" [27] (pp. 140–144)

Creating digital and other forums to discuss normative information and provide mutual support [85]
Providing factual and context information, and tips and guides for action [37,135]

Generating discussions between participants about any aspects of the target behavior and change, including but not limited to the normative ones [35,98,129]

Implementing training and skills building processes that support the target behavior [35,93,136]

Setting goals or creating commitments to action (such as contracts or symbolic vaccination activities [6,95])

Providing financial incentives or penalties compatible with the target behavior [6,21,137]

In special cases, providing psychological therapy [96]

## 5. Limitations and Further Research

As with all literature reviews, our study has several limitations that would benefit from further research efforts. First, regarding our search design and inclusion criteria, although we used broad terms for our search (or because of this), we also had to choose to limit our search to the title of articles only. This was a methodological choice to get a varied yet manageable sample, but further research could include abstracts in the search (as many systematic reviews do [96] but which in our sample yielded more than 40,000 initial hits), as well as limiting thematically, methodologically or in some other way the sample. This meant that our sample left out some landmark studies that did not strictly fulfil our search criteria, and might also explain (although that is debatable) why our final sample had a large representation of similar studies (i.e. computer-delivered health interventions targeted at students, often North American). This bias, already highlighted above, is inherent to the published literature and likely reflects what type of intervention studies get more easily published. Nevertheless, it remains a limitation. Also, because of the difficulty of getting full digital texts and because of the resources we had, we chose to only select studies which had an English language abstract and that were not in books, dissertations, posters or research protocols. We also did not conduct searches of grey literature at this time, which undoubtedly contains very rich and relevant information.

Although we acknowledge that all these decisions limited the representativeness of the studies in our sample among the total population, it did allow us to obtain a diverse yet manageable sample to fulfil the objectives of our in-depth study of methods. Furthermore, we also completed our sample by conducting backward and forward searches, and we contacted expert and relevant authors to enquire for unpublished studies. Our sample, in this sense, did allow us to get a good grasp on how the elements of interest are used in this kind of interventions, and we do not think that the central tendencies we found are radically different to those of the general set of comparable interventions. Other researchers, depending on their objectives, should reflect on the best arrangements to balance the always problematic relationship between available resources, representativeness and variety of the sample, all according to their needs.

Another limitation in our review comes from the amount of information available in the studies. Many articles lacked sufficient details in their procedures for us to tease apart clearly the intervention

procedures, messages (including, for example, if descriptive or injunctive norms were used) and context they used. In this respect, details that are rarely reported in academic articles (such as the specific messages used, the content of workshops and face-to-face interactions, or the times and places of some interventions), are crucial to fully understand how these elements were used in each case. A further study could indeed explore this in more detail by generating surveys for researchers or collecting detailed research protocols.

Finally, the logical next step is to research how the manipulation of the elements of interests in our study can be related to the effects of interventions on behavior. This, for us, is a second and more complicated step that requires a careful design and a different sample to the one we have used. Such a sample would have to manage the difficulties derived from issues such as the quality of the selected studies, their internal and external validity, and the great variety of target behaviors, types of participants, contexts and measures social norm interventions in the literature display, as well as the lack of detailed information about intervention procedures and their mixed use of different strategies and mechanisms. This could be done both through meta-analyses or field experiments, but would likely require a greater focus on specific areas, target behaviors and even dimensions and mechanisms. In our case, as already argued, we opted for a broader and more varied sample that would first allow us to describe and understand some of the dimensions involved and their associations.

## 6. Conclusions

In this paper, we have collected a wide systematic sample of empirical studies to explore the practical strategies and mechanisms through which researchers have leveraged normative information to achieve behavioral changes in real-world contexts. Specifically, we described key differences in terms of the context of application of the interventions, the types of normative information leveraged, and the physical, psychological and social mechanisms used to support the changes. Mainly, we highlighted two strategic dimensions in intervention design using social norms: situated or remote, and transmitting normative information through group summary information or exposure to behaviors and opinions. And based in our results, we also provided some of the elements that should be taken into account to design more integral and sustainable interventions.

Overall, we have underlined and argued in favor of the wide range of ways in which normative information can be leveraged to change behavior in the real world. Indeed, while some interventions and procedures are clearly easier to implement and test in academic studies (and to publish), if we want to advance the practical, real-world effectiveness of social-norm interventions we have to make an effort to explore more diverse and complex interventions (such as the ones done in a situated manner and using EBO), and to improve the way we report and measure the behavioral impact of our interventions. That is because in the real world of action, behavior is determined not just by the agent's will, but also by three layers of determinants in the socially constructed "installation" that funnels and scaffolds behavior: the agents' beliefs and competences indeed, but also the context's material affordances and the social control [27]. The closer to the installation the intervention is, the more efficient it is. Famous photographer Robert Capa used to say: "if your photographs aren't good enough, you're not close enough" [138]. The same seems to apply to intervention: the closest to the actual action, and the closest to the social pressure, the better.

Through this paper, we hope to contribute to the general understanding of how these elements have been and can be included in real-world behavioral change interventions based on social norms, as well as their relevance both to research and practice. It is our hope that by extending our understanding of these factors, we will inform the application of more effective and sustainable interventions to tackle some of the collective challenges that societies and environments all over the world are facing. For practitioners and administrators everywhere, this can mean tapping into a huge potential for much needed real-world change.

**Supplementary Materials:** The following are available online at www.mdpi.com/xxx/s1; Table S1: Full coded set of studies, Figure S1: MCA Output, Figure S2: MCA coordinate plot.

**Author Contributions:** Conceptualization, P.Y. and S.L.; Methodology, P.Y.; Investigation and data curation, P.Y., M.F. and S.L.; Formal analysis, P.Y. and S.L.; Writing—original draft preparation, P.Y.; Writing—review and editing, P.Y. and S.L.

**Funding:** This research project is possible thanks to the generous support of the International Labour Organization (Geneva Headquarters) as part of the "Practical Methods to Change Social Norms in Domestic Work" report by Paulius Yamin and Claire Hobden.

**Acknowledgments:** We would like to extend our gratitude to the colleagues who provided valuable ideas, insights, comments and technical assistance to this research, although they might not agree with all the methodological choices and conclusions of the paper. Of course, any errors are not their responsibility. Thank you to Anam Parand, Mark Noort, Claire Hobden, Maximilian Heitmayer, Kavita Abraham-Dowsing, Juan Pablo Caicedo, Diana Franco and Margaret Tankard. We are also grateful to the reviewers for their comments, which were very useful to further improve the paper.

**Conflicts of Interest:** The authors declare no conflict of interest. The funder had no role in the design of this study; in the collection, analyses, or interpretation of data; in the writing of the manuscript, or in the decision to publish the results.

## Appendix A

**Table A1.** List of studies selected for the review.

| No. | Reference | Author(s) | Year | Title | Country | Intervention topic |
|-----|-----------|-----------|------|-------|---------|--------------------|
| 1 | [94] | Agha and Van Rossem | 2004 | Impact of a school-based peer sexual health intervention on normative beliefs, risk perceptions, and sexual behavior of Zambian adolescents. | Zambia | Sexual health/HIV |
| 2 | [95] | Balvig and Holmberg | 2011 | The ripple effect: A randomized trial of a social norms intervention in a Danish middle school setting. | Denmark | Health behavior |
| 3 | [33] | Bateson et al. | 2013 | Do images of "watching eyes" induce behaviour that is more pro-social or more normative? A field experiment on littering. | UK | Pro-environmental |
| 4 | [137] | Bauer et al. | 2017 | Financial Incentives Beat Social Norms : A Field Experiment on Retirement Information Search. | Netherlands | Tax and retirement |
| 5 | [139] | Bertholet et al. | 2016 | Are young men who overestimate drinking by others more likely to respond to an electronic normative feedback brief intervention for unhealthy alcohol use? | Switzerland | Alcohol |
| 6 | [37] | Bewick et al. | 2013 | The effectiveness of a Web-based | UK | Alcohol |

| No. | Reference | Author(s) | Year | Title | Country | Intervention topic |
|-----|-----------|-----------|------|-------|---------|--------------------|
| | | | | personalized feedback and social norms alcohol intervention on United Kingdom university students: Randomized controlled trial. | | |
| 7 | [109] | Bewick et al. | 2008 | The feasibility and effectiveness of a web-based personalised feedback and social norms alcohol intervention in UK university students: A randomised control trial. | UK | Alcohol |
| 8 | [140] | Boen et al. | Unpublished | Portraying role models to promote stair climbing in a public setting: The effect of matching sex and age | Germany | Health behavior |
| 9 | [88] | Boyle et al. | 2017 | PNF 2.0? Initial evidence that gamification can increase the efficacy of brief, web-based personalized normative feedback alcohol interventions. | U.S. | Alcohol |
| 10 | [135] | Brent et al. | 2017 | Are Normative Appeals Moral Taxes? Evidence from A Field Experiment on Water Conservation | U.S. | Pro-environmental |
| 11 | [136] | Bruce and Keller | 2007 | Applying Social Norms Theory within groups: Promising for high-risk drinking. | U.S. | Alcohol |
| 12 | [80] | Brudermann et al. | 2017 | Eyes on social norms: A field study on an honor system for newspaper sale. | Austria | Integrity - morality |
| 13 | [40] | Celio and Lisman | 2014 | Examining the efficacy of a personalized normative feedback intervention to reduce college student gambling. | U.S. | Gambling |
| 14 | [38] | Chernoff and Davison | 2005 | An evaluation of a brief HIV/AIDS prevention intervention for college students | U.S. | Sexual health / HIV |

| No. | Reference | Author(s) | Year | Title | Country | Intervention topic |
|-----|-----------|-----------|------|-------|---------|--------------------|
| | | | | using normative feedback and goal setting. | | |
| 15 | [141] | Cleveland et al. | 2013 | Moderation of a parent-based intervention on transitions in drinking: Examining the role of normative perceptions and attitudes among high- and low-risk first-year college students. | U.S. | Alcohol |
| 16 | [142] | Collins et al. | 2002 | Mailed personalized normative feedback as a brief intervention for at-risk college drinkers. | U.S. | Alcohol |
| 17 | [143] | Collins et al. | 2014 | Randomized controlled trial of web-based decisional balance feedback and personalized normative feedback for college drinkers. | U.S. | Alcohol |
| 18 | [87] | Cross and Peisner | 2009 | RECOGNIZE: A social norms campaign to reduce rumor spreading in a junior high school. | U.S. | Bullying / harassment behavior |
| 19 | [144] | Cunningham and Wong | 2013 | Assessing the immediate impact of normative drinking information using an immediate post-test randomized controlled design: Implications for normative feedback interventions? | Canada | Alcohol |
| 20 | [96] | Davis et al. | 2016 | Brief Motivational Interviewing and Normative Feedback for Adolescents: Change Language and Alcohol Use Outcomes. | U.S. | Alcohol |
| 21 | [145] | Dejong et al. | 2006 | A multisite randomized trial of social norms marketing campaigns to reduce college student drinking: A replication failure. | U.S. | Alcohol |
| 22 | [146] | Doumas and Hannah | 2008 | Preventing high-risk drinking in | U.S. | Alcohol |

| No. | Reference | Author(s) | Year | Title | Country | Intervention topic |
|---|---|---|---|---|---|---|
| | | | | youth in the workplace: A web-based normative feedback program. | | |
| 23 | [147] | Doumas et al. | 2010 | Reducing heavy drinking among first year intercollegiate athletes: A randomized controlled trial of web-based normative feedback. | U.S. | Alcohol |
| 24 | [148] | Doumas et al. | 2011 | Reducing high-risk drinking in mandated college students: Evaluation of two personalized normative feedback interventions. | U.S. | Alcohol |
| 25 | [17] | Flüchter et al. | 2014 | Digital commuting: The effect of social normative feedback on e-bike commuting - Evidence from a field study. | Switzerland | Pro-environmental |
| 26 | [36] | Gidycz et al. | 2011 | Preventing sexual aggression among college men: An evaluation of a social norms and bystander intervention program. | U.S. | Bullying / harassment behavior |
| 27 | [64] | Hallsworth et al. | 2016 | Provision of social norm feedback to high prescribers of antibiotics in general practice: a pragmatic national randomised controlled trial. | UK | Health behavior |
| 28 | [149] | Hartwell and Campion | 2016 | Getting on the same page: The effect of normative feedback interventions on structured interview ratings. | U.S. | Rating of interviews |
| 29 | [19] | Hopper et al. | 1991 | Recycling as altruistic behavior: Normative and Behavioral Strategies to Expand Participation in a Community Recycling Program. | U.S. | Pro-environmental |
| 30 | [150] | Howe et al. | Unpublished | Normative Appeals Are More Effective When They Invite People | U.S. | Pro-environmental |

| No. | Reference | Author(s) | Year | Title | Country | Intervention topic |
|-----|-----------|-----------|------|-------|---------|--------------------|
| | | | | to Work Together Toward a Common Cause | | |
| 31 | [93] | Kearney et al. | 2013 | The impact of an alcohol education program using social norming. | U.S. | Alcohol |
| 32 | [151] | Koeneman et al. | 2017 | A novel method to promote physical activity among older adults in residential care: An exploratory field study on implicit social norms. | Netherlands | Health behavior |
| 33 | [16] | Kormos et al. | 2015 | The Influence of Descriptive Social Norm Information on Sustainable Transportation Behavior: A Field Experiment. | Canada | Pro-environmental |
| 34 | [152] | Kulik et al. | 2008 | Social norms information enhances the efficacy of an appearance-based sun protection intervention. | U.S. | Health behavior |
| 35 | [153] | Labrie et al. | 2009 | A brief live interactive normative group intervention using wireless keypads to reduce drinking and alcohol consequences in college student athletes. | U.S. | Alcohol |
| 36 | [85] | Lahlou et al.. | 2015 | Increasing water intake of children and parents in the family setting: a randomized, controlled intervention using installation theory. | Poland | Water intake |
| 37 | [24] | Lapinski et al. | 2013 | Testing the Effects of Social Norms and Behavioral Privacy on Hand Washing: A Field Experiment. | U.S. | Health behavior |
| 38 | [90] | Latkin et al. | 2013 | The dynamic relationship between social norms and behaviors: the results of an HIV prevention network intervention for injection drug users. | U.S. | Sexual health / HIV |
| 39 | [154] | Lewis, and Neighbors | 2006 | An Examination of College Student Activities and Attentiveness | U.S. | Alcohol |

| No. | Reference | Author(s) | Year | Title | Country | Intervention topic |
|-----|-----------|-----------|------|-------|---------|--------------------|
| | | | | During a Web-Delivered Personalized Normative Feedback Intervention. | | |
| 40 | [119] | Lewis et al. | 2008 | 21st Birthday Celebratory Drinking: Evaluation of a Personalized Normative Feedback Card Intervention. | U.S. | Alcohol |
| 41 | [132] | Lewis et al. | 2014 | Randomized controlled trial of a web-delivered personalized normative feedback intervention to reduce alcohol-related risky sexual behavior among college students. | U.S. | Alcohol |
| 42 | [155] | Lojewski et al. | 2010 | Personalized normative feedback to reduce drinking among college students: A social norms intervention examining gender-based versus standard feedback. | U.S. | Alcohol |
| 43 | [156] | Mattern and Neighbors | 2004 | Social norms campaigns: Examining the relationship between changes in perceived norms and changes in drinking levels. | U.S. | Alcohol |
| 44 | [39] | McCoy et al. | 2017 | Pilot study of a multi-pronged intervention using social norms and priming to improve adherence to antiretroviral therapy and retention in care among adults living with HIV in Tanzania. | Tanzania | Sexual health / HIV |
| 45 | [6] | Mockus | 2002 | Co-existence as harmonization of law, morality and culture. | Colombia | Pro-environmental—Use of pedestrian crossings |
| 46 | [92] | Mogro-Wilson et al. | 2017 | A Brief High School Prevention Program to Decrease Alcohol Usage and Change Social Norms. | U.S. | Alcohol |
| 47 | [20] | Mollen et al. | 2013 | Healthy and unhealthy social norms and food | U.S. | Health behavior |

| No. | Reference | Author(s) | Year | Title | Country | Intervention topic |
|-----|-----------|-----------|------|-------|---------|--------------------|
| | | | | selection. Findings from a field-experiment. | | |
| 48 | [157] | Mollen et al. | 2013 | Intervening or interfering? The influence of injunctive and descriptive norms on intervention behaviours in alcohol consumption contexts. | Netherlands | Alcohol |
| 49 | [84] | Moore et al. | 2013 | An exploratory cluster randomised trial of a university halls of residence based social norms marketing campaign to reduce alcohol consumption among 1st year students. | UK | Alcohol |
| 50 | [158] | Moreira et al. | 2012 | Personalised Normative Feedback for Preventing Alcohol Misuse in University Students: Solomon Three-Group Randomised Controlled Trial. | UK | Alcohol |
| 51 | [89] | Neighbors et al. | 2004 | Targeting misperceptions of descriptive drinking norms: Efficacy of a computer-delivered personalized normative feedback intervention. | U.S. | Alcohol |
| 52 | [159] | Neighbors et al. | 2010 | Efficacy of web-based personalized normative feedback: A two-year randomized controlled trial. | U.S. | Alcohol |
| 53 | [160] | Neighbors et al. | 2011 | Social-norms interventions for light and nondrinking students. | U.S. | Alcohol |
| 54 | [41] | Neighbors et al. | 2015 | Efficacy of personalized normative feedback as a brief intervention for college student gambling: A randomized controlled trial. | U.S. | Gambling |
| 55 | [161] | Neighbors et al. | 2006 | Being controlled by normative influences: Self- | U.S. | Alcohol |

| No. | Reference | Author(s) | Year | Title | Country | Intervention topic |
|---|---|---|---|---|---|---|
| | | | | determination as a moderator of a normative feedback alcohol intervention. | | |
| 56 | [35] | Paluck and Shepherd H. | 2012 | The salience of social referents: A field experiment on collective norms and harassment behavior in a school social network. | U.S. | Bullying / harassment behavior |
| 57 | [22] | Payne et al. | 2015 | Shopper marketing nutrition interventions: Social norms on grocery carts increase produce spending without increasing shopper budgets. | U.S. | Health behavior |
| 58 | [162] | Pedersen et al. | 2016 | A randomized controlled trial of a web-based, personalized normative feedback alcohol intervention for young-adult veterans. | U.S. | Alcohol |
| 59 | [14] | Pellerano et al. | 2016 | Do Extrinsic Incentives Undermine Social Norms? Evidence from a Field Experiment in Energy Conservation. | Ecuador | Pro-environmental |
| 60 | [129] | Perkins and Craig | 2006 | A successful social norms campaign to reduce alcohol misuse among college student-athletes. | U.S. | Alcohol |
| 61 | [163] | Perkins et al. | 2011 | Using social norms to reduce bullying: A research intervention among adolescents in five middle schools. | U.S. | Bullying / harassment behavior |
| 62 | [111] | Perkins et al. | 2010 | Effectiveness of social norms media marketing in reducing drinking and driving: A statewide campaign. | U.S. | Alcohol |
| 63 | [25] | Pfattheicher et al. | Unpublished | A Field Study on Watching Eyes and Hand Hygiene Compliance in a Public Restroom | Germany | Health behavior |
| 64 | [98] | Prince et al. | 2015 | Development of a Face-to-Face Injunctive Norms Brief Motivational | U.S. | Alcohol |

| No. | Reference | Author(s) | Year | Title | Country | Intervention topic |
|---|---|---|---|---|---|---|
| | | | | Intervention for College Drinkers and Preliminary Outcomes. | | |
| 65 | [164] | Reid and Aiken | 2013 | Correcting injunctive norm misperceptions motivates behavior change: A randomized controlled sun protection intervention. | U.S. | Health behavior |
| 66 | [165] | Reilly and Wood | 2008 | A randomized test of a small-group interactive social norms intervention. | U.S. | Alcohol |
| 67 | [166] | Ridout and Campbell | 2014 | Using facebook to deliver a social norm intervention to reduce problem drinking at university. | Australia | Alcohol |
| 68 | [167] | Schultz and Tyra | Unpublished | Two Field Studies of Normative Beliefs and Environmental Behavior | U.S. | Pro-envirnomental |
| 69 | [15] | Schultz et al. | 2016 | Personalized normative feedback and the moderating role of personal norms: A field experiment to reduce residential water consumption. | U.S. | Pro-environmental |
| 70 | [168] | Schultz et al. | Unpublished | Normative Social Influence Transcends Culture, But Detecting It Is Culture Specific | Multiple | Alcohol |
| 71 | [18] | Schultz | 1999 | Changing behavior with normative feedback interventions: A field experiment on curbside recycling. | U.S. | Pro-envirnomental |
| 72 | [169] | Scribner et al. | 2011 | Alcohol prevention on college campuses: The moderating effect of the alcohol environment on the effectiveness of social norms marketing campaigns. | U.S. | Alcohol |
| 73 | [130] | Silk et al.. | 2017 | Evaluation of a Social Norms Approach to a Suicide Prevention Campaign. | U.S. | Counselling service use |
| 74 | [131] | Smith et al. | 2006 | A social judgment theory approach to | U.S. | Alcohol |

| No. | Reference | Author(s) | Year | Title | Country | Intervention topic |
|-----|-----------|-----------|------|-------|---------|--------------------|
| | | | | conducting formative research in a social norms campaign. | | |
| 75 | [124] | Spijkerman et al. | 2010 | Effectiveness of a Web-based brief alcohol intervention and added value of normative feedback in reducing underage drinking: A randomized controlled trial. | U.S. | Alcohol |
| 76 | [170] | Stamper et al. | 2004 | Replicated findings of an evaluation of a brief intervention designed to prevent high-risk drinking among first-year college students: Implications for social norming theory. | U.S. | Alcohol |
| 77 | [171] | Steffian | 1999 | Correction of normative misperceptions: An alcohol abuse prevention program. | U.S. | Alcohol |
| 78 | [172] | Su et al. | 2017 | Evaluating the Effect of a Campus-wide Social Norms Marketing Intervention on Alcohol Use Perceptions, Consumption, and Blackouts. | U.S. | Alcohol |
| 79 | [173] | Taylor et al. | 2015 | Improving social norms interventions: Rank-framing increases excessive alcohol drinkers' information-seeking. | UK | Alcohol |
| 80 | [174] | Thombs et al. | 2007 | Outcomes of a technology-based social norms intervention to deter alcohol use in freshman residence halls. | U.S. | Alcohol |
| 81 | [127] | Thombs et al. | 2004 | A close look at why one social norms campaign did not reduce student drinking. | U.S. | Alcohol |
| 82 | [21] | Thorndike et al. | 2016 | Social norms and financial incentives to promote employees' healthy food choices: A | U.S. | Health behavior |

| No. | Reference | Author(s) | Year | Title | Country | Intervention topic |
|---|---|---|---|---|---|---|
| | | | | randomized controlled trial. | | |
| 83 | [134] | Toghianifar et al. | 2014 | Women's attitude toward smoking: effect of a community-based intervention on smoking-related social norms. | Iran | Health behavior |
| 84 | [133] | Turner et al. | 2008 | Declining negative consequences related to alcohol misuse among students exposed to a social norms marketing intervention on a college campus. | U.S. | Alcohol |
| 85 | [81] | Wall and Cameron | 2017 | Trial of Social Norm Interventions to Increase Physical Activity. | U.S. | Health behavior |
| 86 | [175] | Wegs et al. | 2016 | Community Dialogue to Shift Social Norms and Enable Family Planning: An Evaluation of the Family Planning Results Initiative in Kenya. | Kenya | Sexual health/HIV |
| 87 | [176] | Wenzel | 2005 | Misperceptions of social norms about tax compliance: From theory to intervention. | Australia | Tax and retirement |
| 88 | [120] | Werch et al. | 2000 | Results of a social norm intervention to prevent binge drinking among first-year residential college students. | U.S. | Alcohol |
| 89 | [91] | Yurasek et al. | 2015 | Descriptive Norms and Expectancies as Mediators of a Brief Motivational Intervention for Mandated College Students Receiving Stepped Care for Alcohol Use. | U.S. | Alcohol |

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
