# Peer review of "Using Social Norms to Change Behavior and Increase Sustainability in the Real World: a Systematic Review of the Literature"

_sustainability, doi:10.3390/su11205847_

Round 1
Reviewer 1 Report
This manuscript reviews behavioral change interventions based on social norms and proposes a framework distinguishing between intervention contexts (i.e., situated and remote) and type of normative information (i.e., summary information about a group and exposure to others’ opinions and behaviors). Practical mechanisms and recommendations for policy applications are discussed. This is an interesting topic with great practical relevance, and I am happy to review the manuscript.
However, there are a couple of minor issues I'd kindly ask the authors to consider in their revisions.
Most importantly, I believe that the paper could have a much bigger impact and guide future interventions with higher quality if effect sizes were reported and, where possible, compared. I am aware that this is difficult because the studies differ in many crucial aspects. However, including the effect sizes would give readers an idea of what could be expected by social norm interventions.
The paper is well-written and clearly structured. The figures and tables are helpful and add to readers’ understanding. However, I wonder what the footnote 1 in figure 3 (EBO) means and whether the authors could add the number of studies and participants to their groups 1 and 2 in table 4. Also, the formatting of table 6 makes it hard to read and distinguish between major and minor bullet points. Adding the number of participants, studies and the study design (for example whether there was a control group or not) to table A1 would enrich the overview.
Author Response
Thank you very much for your insightful comments to our manuscript, which have been very useful to improve it further.
Please find attached the cover letter with the answers to your comments and the updated manuscript.

Reviewer 2 Report
Please see attachment.

Author Response

(The authors gave the same response as above.)
